# Decoupled charge and heat transport in Fe$_2$VAl composite thermoelectrics with topological-insulating grain boundary networks

Fabian Garmroudi [1,2] ✉, Illia Serhiienko [2], Michael Parzer [1], Sanyukta Ghosh[3], Pawel Ziolkowski [3], Gregor Oppitz[3], Hieu Duy Nguyen[4], Cédric Bourgès [2,5], Yuya Hattori [2], Alexander Riss [1], Sebastian Steyrer[1], Gerda Rogl [6], Peter Rogl[6], Erhard Schafler[7], Naoyuki Kawamoto[4], Eckhard Müller [3,8], Ernst Bauer[1], Johannes de Boor [3,9] ✉ & Takao Mori [2,10] ✉

Decoupling charge and heat transport is essential for optimizing thermoelectric materials. Strategies to inhibit lattice-driven heat transport, however, also compromise carrier mobility, limiting the performance of most thermoelectrics, including Fe$_2$VAl Heusler compounds. Here, we demonstrate an innovative approach, which bypasses this tradeoff: via liquid-phase sintering, we incorporate the archetypal topological insulator Bi$_{1-x}$Sb$_x$ between Fe$_2$V$_{0.95}$Ta$_{0.1}$Al$_{0.95}$ grains. Structural investigations alongside extensive thermoelectric and magneto-transport measurements reveal distinct modifications in the microstructure, a reduced lattice thermal conductivity and a simultaneously enhanced carrier mobility arising from topologically protected charge transport along the grain boundaries. This yields a huge performance boost, resulting in one of the highest figure of merits among both half- and full-Heusler compounds, $z \approx 1.6 \times 10^{-3}$ K$^{-1}$ ($zT \approx 0.5$) at 295 K. Our findings highlight the potential of topological-insulating secondary phases to decouple charge and heat transport and call for more advanced theoretical studies of multiphase composites.

Given the increasing global demand for efficient energy utilization, thermoelectrics (TEs) present a promising solution as they can harvest decentralized waste heat sources or function as Peltier coolers, e.g., for thermal management applications[1,2]. The conversion efficiency of TE devices depends on the hot- and cold-side temperatures and a material-dependent figure of merit, $z \propto \mu_W/\kappa_L$. The highest achievable $z$ in a semiconductor with optimized carrier concentration is determined by the weighted carrier mobility $\mu_W$ of electrons or holes, which should be maximized, and by the lattice thermal conductivity $\kappa_L$, which should be minimized[3,4]. The inherent tradeoff between $\mu_W$ and $\kappa_L$

[1]Institute of Solid State Physics, TU Wien, Vienna, Austria. [2]International Center for Materials Nanoarchitectonics (WPI-MANA), National Institute for Materials Science (NIMS), Tsukuba, Japan. [3]Institute of Materials Research, German Aerospace Center (DLR), Cologne, Germany. [4]Center for Basic Research on Materials (CBRM), National Institute for Materials Science (NIMS), Tsukuba, Japan. [5]International Center for Young Scientists, National Institute for Materials Science (NIMS), Tsukuba, Japan. [6]Institute of Materials Chemistry, University of Vienna, Vienna, Austria. [7]Faculty of Physics, University of Vienna, Vienna, Austria. [8]Institute of Inorganic and Analytical Chemistry, Justus Liebig University Giessen, Giessen, Germany. [9]University of Duisburg-Essen, Faculty of Engineering, Institute of Technology for Nanostructures (NST) and CENIDE, Duisburg, Germany. [10]Graduate School of Pure and Applied Sciences, University of Tsukuba, Tsukuba, Japan. ✉e-mail: f.garmroudi@gmx.at; Johannes.deBoor@dlr.de; mori.takao@nims.go.jp

presents one of the most formidable challenges in the design and optimization of TE materials, requiring the decoupling of charge and heat transport, that is, the realization of a phonon-glass electron-crystal concept.

Since the mid-20th century, $Bi_2Te_3$-based systems have been the gold standard for TEs operating near room temperature, and currently, they remain the only commercially available option[5,6]. However, the scarcity of tellurium, along with the brittle nature and poor mechanical properties of these materials limits their widespread use in everyday life and industrial applications. Therefore, it is crucial to explore alternatives that offer competitive performance and overcome the challenges related to $Bi_2Te_3$.

For *n*-type materials, cost-effective $Mg_3(Bi,Sb)_2$ Zintl compounds have been considered the hottest candidates as they exhibit very high $z$[7–9]. However, these materials, especially the Bi-rich alloys with attractive near-room temperature properties, suffer from poor chemical stability and degrade rapidly when exposed to air, presenting an ongoing challenge for practical applications.

On the other hand, Heusler compounds based on $Fe_2VAl$, the focus of this study, exhibit excellent chemical and mechanical stability. These materials are also composed of earth-abundant, inexpensive elements with great recyclability[10] – sustainability aspects that are becoming increasingly important worldwide, and particularly within the EU. Moreover, they display outstanding electronic transport properties, with weighted mobilities that are comparable to or even greater than those of other state-of-the-art TEs[11]. Yet, their intrinsically large $\kappa_L$ limits their potential as TE materials[12]. Consequently, previous studies have primarily focused on reducing $\kappa_L$ by substituting heavy elements[13–15], lowering the dimensionality through thin-film deposition[16–18], or reducing the grain size[12,19,20]. Although these strategies have resulted in enhancements of $z$, the overall performance remains a significant bottleneck and is too low for most practical applications.

In this study, we demonstrate that by incorporating chemically and structurally distinct $Bi_{1-x}Sb_x$ at the grain boundaries, charge, and heat transport can be decoupled, resulting in a reduction of $\kappa_L$, and simultaneously, in an unexpected increase of $\mu_W$ (Fig. 1a). Consequently, the figure of merit is boosted by more than a factor of two, up to $z_{max} \approx 1.7 \times 10^{-3}\ K^{-1}$ at 240–250 K ($z \approx 1.6 \times 10^{-3}\ K^{-1}$ at room

temperature), representing one of the largest values hitherto reported among *n*-type half- and full-Heusler compounds (Fig. 1b).

## Results

### Decoupling charge and heat transport in $Fe_2VAl$

The lattice thermal conductivity of $Fe_2VAl$ Heusler compounds is intrinsically large, $\kappa_L \approx 27\ W\ m^{-1}\ K^{-1}$ at 300 K[21], which can be mainly attributed to a lack of structural and chemical bonding complexity as well as the absence of heavy elements, leading to high sound velocities. Upon alloying, $\kappa_L$ can be drastically reduced down to $10\ W\ m^{-1}\ K^{-1}$ in $Fe_2VAl_{1-x}Si_x$[21], $7\ W\ m^{-1}\ K^{-1}$ in $Fe_2VAl_{1-x}Ge_x$[21], and by substituting heavy $5d$ elements, further down to $5\ W\ m^{-1}\ K^{-1}$ in $Fe_2VTa_xAl_{1-x}$[14] and $4\ W\ m^{-1}\ K^{-1}$ in $Fe_2V_{1-x}W_xAl$[15]. As a downside, the very same point defects, which effectively inhibit heat transport by high-frequency phonons, also strongly scatter charge carriers. This is particularly true for the $5d$ elements like Ta and W, which are substituted for V atoms. Since V-$d$ states dominate the electronic states of the conduction band, introducing substitutional disorder at the V site results in intense electronic scattering. We gathered TE property data from various substitution studies[13,14,21–23] and calculated $\mu_W$[4]. A strong tradeoff relationship between $\mu_W$ and $\kappa_L$ is obvious from Fig. 1a (black symbols).

Aside from introducing point defects, $\kappa_L$ can be suppressed by reducing the grain size $d$ and several studies attempted to enhance $z$ by grain size reduction, e.g., via ball milling[13,19] or high-pressure torsion (HPT)[20,24], yielding $d \approx 100\ nm$. Employing HPT, Fukuta et al. recently reported very low values of $\kappa_L$ down to $1.3\ W\ m^{-1}\ K^{-1}$ in $Fe_2V_{0.98}Ta_{0.1}Al_{0.92}$ at 350 K and $zT$ up to 0.37 at 400 K ($z \approx 0.9 \times 10^{-3}\ K^{-1}$)[24]. These remarkable findings motivated us to (i) reproduce them and (ii) apply HPT to a variety of different samples with optimized compositions. The results of these endeavors are summarized in the Supplementary Information (SI). While $\kappa_L$ could indeed be dramatically reduced down to $<2\ W\ m^{-1}\ K^{-1}$, we concomitantly observed a huge deterioration of electronic transport in all cases (see Figs. S1 and S2 and blue symbols in Fig. 1a), resulting in no enhancement of $zT$ (Fig. S3). Similar observations have been made, e.g., for $Mg_3(Bi,Sb)_2$[25,26] and various half-Heuslers, where reducing grain size comes at a cost of reducing $\mu_W$[27,28]. The discrepancy between our results and previous ones from ref. 29 suggests that setup-specific conditions during HPT are generally very important, complicating reproducibility and upscale production.

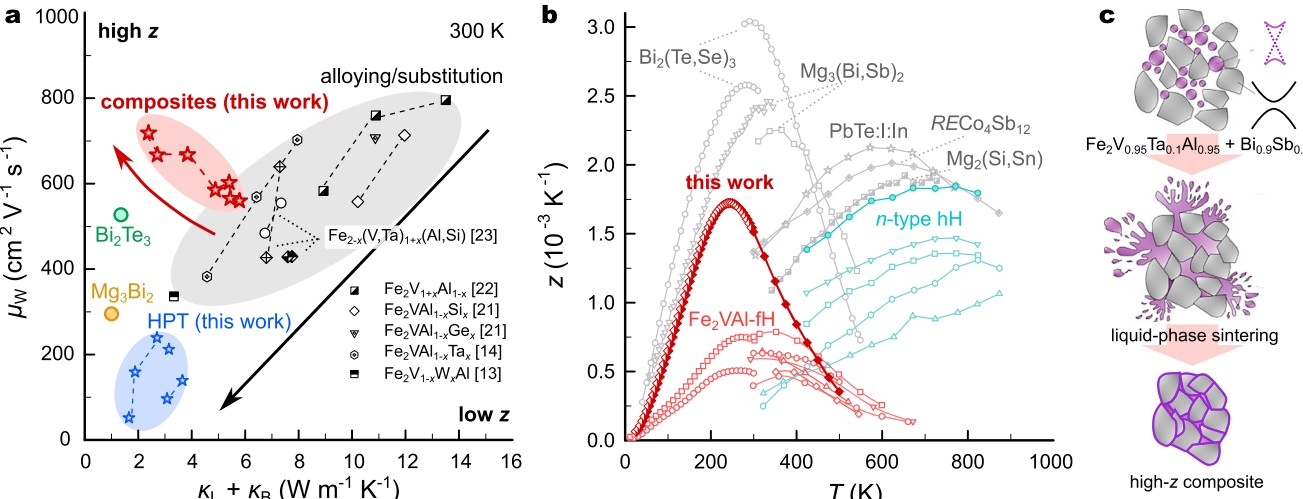

**Fig. 1 | Boosting thermoelectric performance in Heusler compounds by decoupled charge and heat transport. a** Tradeoff between weighted mobility and lattice thermal conductivity (plus bipolar term $\kappa_B$) in $Fe_2VAl$-based Heusler compounds at room temperature[13,14,21–23]. Data for state-of-the-art *n*-type $Bi_2Te_3$- and $Mg_3Bi_2$-based systems[6,7] at 300 K are shown for comparison. Composites in this

work are found to bypass this tradeoff. **b** Temperature-dependent figure of merit of the best composite from this work (FVAB50), compared to other optimally doped, high-performance *n*-type thermoelectrics[50–55], reaching one of the highest $z$ among both half-Heusler (hH) and full-Heusler (fH) compounds. **c** Schematic synthesis of composites via liquid-phase sintering.

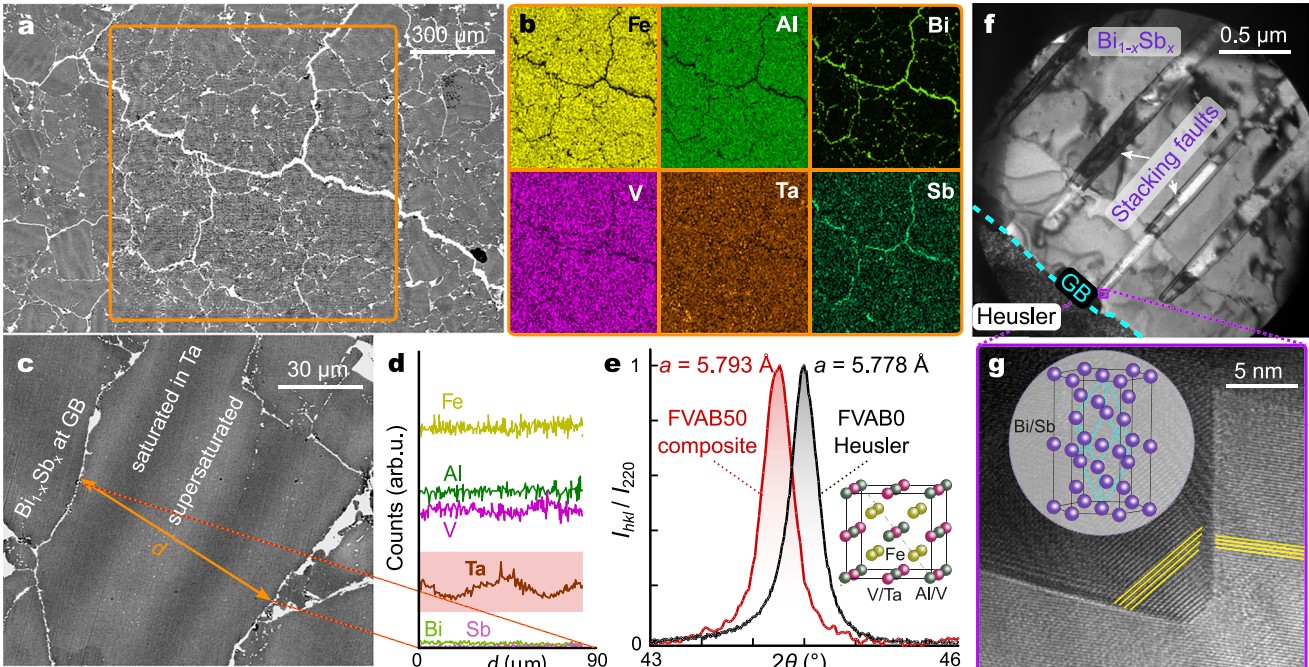

**Fig. 2 | Microstructure evolution in $Fe_2V_{0.95}Ta_{0.1}Al_{0.95}$ Heusler compounds upon incorporating $Bi_{1-x}Sb_x$. a** Microstructure of FVAB50 composite, where the majority of GBs are filled with Bi-Sb. **b** EDX analyses reveal that Bi and Sb are found at the GBs, while Fe, V, Al, and Ta are almost exclusively distributed within the grains. **c** BS-SEM image of a Heusler grain with periodic contrast variations, surrounded by Bi-Sb. **d** EDX line scan along the Heusler grain shown in (**c**). **e** Comparison of normalized X-ray diffraction peaks of the (220) plane of $Fe_2V_{0.95}Ta_{0.1}Al_{0.95}$ and FVAB50 composite. **f** Bright-field TEM image of the GB with ladder-like nanostructure arrays of stacking faults and (**g**) high-magnification image near stacking-faults. Insets in (**e** and **g**) show Heusler and Bi-Sb unit cells, respectively.

Instead, we have devised a different approach wherein chemically and structurally distinct $Bi_{1-x}Sb_x$ is incorporated as a secondary phase between the Heusler grains. Figure 1c outlines the synthesis procedure. The starting materials were first synthesized using an induction melting furnace and then hand-ground into a fine powder. The powders were mixed in various ratios (5–50 vol.% $Bi_{0.9}Sb_{0.1}$), and sintered at 1373 K. The much lower melting point of $Bi_{0.9}Sb_{0.1}$ causes excess liquid to be expelled during sintering. The retention of $Bi_{1-x}Sb_x$ in the composite depends on the particle size of the Heusler phase and the amount of $Bi_{0.9}Sb_{0.1}$ used. Backscattered scanning electron microscopy (BS-SEM) shows that up to ≈30 vol.%, $Bi_{0.9}Sb_{0.1}$ fills only the triple junctions of Heusler grains, while 50 vol.% $Bi_{0.9}Sb_{0.1}$ allows the liquid phase to wet and coat all grains as a grain boundary (GB) phase (Fig. S10). This produces highly dense ($Fe_2V_{0.95}Ta_{0.1}Al_{0.95}$ + $Bi_{0.9}Sb_{0.1}$) composites, referred to as FVAB$X$, with $X$ indicating the $Bi_{0.9}Sb_{0.1}$ volume percentage added before sintering. The reference sample, without any $Bi_{1-x}Sb_x$, achieved a density of approximately 95% of its theoretical density. When about 10 vol.% $Bi_{0.9}Sb_{0.1}$ are added before sintering, the composite material shows minimal porosity. SEM micrographs confirm that any pores present in the initial sample are filled by the secondary $Bi_{1-x}Sb_x$ phase, resulting in a density close to 100% for all composite samples.

The $\mu_W$ versus $\kappa_L$ trend (see Fig. 1a) for the composite samples is unusual and cardinally different from other approaches, like alloying or grain size reduction via HPT. Moreover, the exceptionally high $\mu_W$, in spite of the suppressed $\kappa_L$, signifies a decoupling of charge and lattice-driven heat transport. In the following, we present investigations of the microstructure of these materials alongside local microscale probing of the Seebeck coefficient $S$. Finally, we show and discuss experimental results from extensive TE and magneto-transport measurements carried out in a broad range of temperatures and magnetic fields.

## Structural modifications in FVAB$X$ composites

The structural properties of the sintered samples were investigated via scanning and transmission electron microscopy (TEM), energy-dispersive X-ray spectroscopy (EDX), and X-ray diffraction (XRD). Fig. S4 shows BS-SEM images of $Fe_2V_{0.95}Ta_{0.1}Al_{0.95}$ sintered at 1373 K without the addition of $Bi_{0.9}Sb_{0.1}$. Throughout the whole sample, nanoscale precipitates of a secondary Ta-rich phase are clearly noticeable at the GBs. This is in agreement with the previously established low solubility limit of Ta, $x = 0.07$ in $Fe_2V_{1-x}Ta_xAl$[30]. Apart from that, the microstructure displays a very homogeneous phase distribution without any variations in the composition.

Figure 2a shows a low-magnification image of the microstructure of the FVAB50 composite, with the best TE properties. A uniform distribution of $Bi_{1-x}Sb_x$ along the GBs is obvious and confirmed by compositional mapping using EDX (Fig. 2b) with an estimated volume fraction of around 5–7 vol. %. Additionally, we find that the segregation of $Bi_{1-x}Sb_x$ along the GBs goes hand in hand with two changes in the microstructure: (i) strong suppression of nanoscale Ta-rich precipitates at the GBs, (ii) diffuse brightness variations within the grains. Both these structural changes suggest an enhanced solubility limit of heavy Ta atoms, when $Bi_{1-x}Sb_x$ is incorporated as a GB network during the liquid-phase sintering, contributing to a reduction of the lattice thermal conductivity as shown later. This is confirmed by EDX line scans (Fig. 2d) across the grain, revealing periodic fluctuations in the Ta and V concentration, and XRD, revealing an increase of the lattice parameter of the Heusler phase (Fig. 2e and inset of Fig. S12c) as larger and heavier Ta atoms are substituted. In Fig. 2f, g, we focus on $Bi_{1-x}Sb_x$. Since $Fe_2VAl$ and $Bi_{1-x}Sb_x$ are chemically and structurally distinct, there exists a well-defined GB without apparent interdiffusion. We find that $Bi_{1-x}Sb_x$, when embedded between Heusler grains, displays a peculiar ladder-like nanostructure with arrays of stacking fault defects. Moreover, detailed EDX analyses of different samples indicate that the Bi:Sb ratio fluctuates and that the Sb concentration is above the nominal one

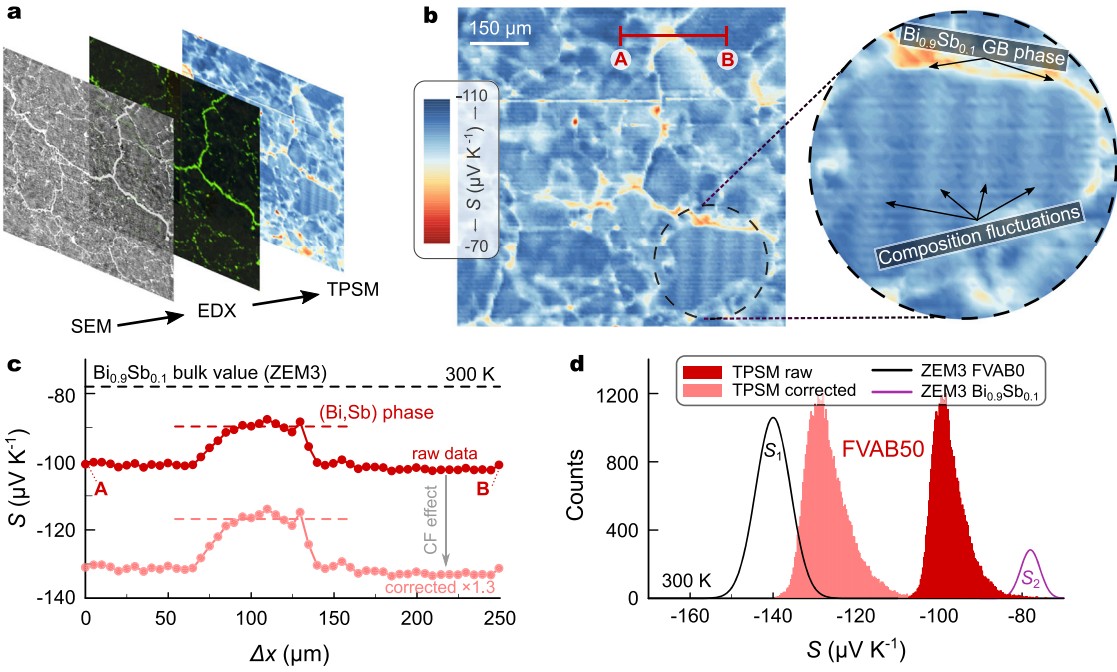

**Fig. 3 | Local investigation of electronic transport in (Fe$_2$V$_{0.95}$Ta$_{0.1}$Al$_{0.95}$ + Bi$_{0.9}$Sb$_{0.1}$) composites. a** Microstructure, composition, and local transport probing in the same area (orange square in Fig. 2a). **b, c** Transient potential Seebeck microprobe (TPSM) mapping of FVAB50 at room temperature. Ta-enriched regions inside the Heusler grains and Bi$_{1-x}$Sb$_x$ at the GBs display a smaller $S$ than the remaining part of the matrix. **c** TPSM line scans across distance marked in (**b**). **d** Histogram from TPSM mapping. Solid lines are normal distributions centered around $S_1 = -140\ \mu V^{-1} K^{-1}$ and $S_2 = -78\ \mu V^{-1} K^{-1}$, the bulk values for Fe$_2$V$_{0.95}$Ta$_{0.1}$Al$_{0.95}$ and Bi$_{0.9}$Sb$_{0.1}$, respectively.

(although remaining within the topological-insulating regime) in our high-performance FVAB50 composite (Figs. S9 and S11).

## Thermoelectric properties

The pivotal role of understanding and investigating TE transport across grain boundaries is increasingly recognized[25,29,31,32]. To draw a connection between microstructure and electronic transport we employed a transient potential Seebeck microprobe (TPSM), with local property investigations performed on the same rectangular area of the sample (Fig. 3a). Figure 3b shows a map of the locally determined $S$ with a spatial resolution of 3–5 microns. The results obtained are in excellent agreement with structural investigations revealing a rich and complex microstructure consisting of Heusler grains and a Bi$_{1-x}$Sb$_x$ GB network. Interestingly, TPSM measurements suggest that Bi$_{1-x}$Sb$_x$ exhibits a larger $S$ as a secondary phase compared to its bulk form. This is emphasized by looking at line scans across Heusler grains. The plateau in Fig. 3c refers to the value of Bi$_{1-x}$Sb$_x$ within the composite, which is significantly higher than its bulk value, especially considering that TPSM measurements typically underestimate $S$ by at least 20–30% due to the cold finger effect. This enhancement, which exceeds the highest $S$ at 300 K in the entire composition range of polycrystalline Bi$_{1-x}$Sb$_x$[33], is surprising, given the near-complete immiscibility between Bi$_{1-x}$Sb$_x$ and the Heusler phase.

Figure 3d shows the distribution histogram of the measured Seebeck coefficient. For the Heusler phase, $S_1 \approx -140\ \mu V K^{-1}$, and for Bi$_{0.9}$Sb$_{0.1}$, $S_2 \approx -78\ \mu V K^{-1}$ would be expected. However, instead of two normal distributions centered around those values, the observed distribution appears much more merged with $S$ being significantly under(over)estimated with respect to $S_1(S_2)$. While the underestimation is an artifact from the cold finger effect, inherent to the TPSM and basically all microprobe measurements[34], the enhanced $S$ of the secondary phase indicates a beneficial interplay between the two components and explains why the integral value of $S$ remains large in the composite (Fig. 4c), despite being short-circuited across the GBs.

We note that a similar observation has been made already several years ago by Mikami and Kobayashi in (Fe$_2$VAl$_{0.9}$Si$_{0.1}$ + Bi) composites with $zT_{max} = 0.11$[35].

In Fig. 4, we compare the temperature-dependent bulk TE properties of our FVAB$X$ ($X$ = 0, 20, 50) composites over a broad temperature range from 4 to 523 K. Measurements were performed using different setups in various laboratories at the National Institute for Materials Science (NIMS) in Japan and at TU Wien (TUW) in Austria. Additionally, extensive measurements have also been conducted on a bulk sample of Bi$_{0.9}$Sb$_{0.1}$ synthesized during this study, and the data have been included for comparison. The latter are in excellent agreement with those reported previously (see Fig. S13).

Figure 4a displays the temperature-dependent thermal conductivity $\kappa(T)$. At 200–300 K, $\kappa(T)$ increases due to bipolar thermal transport, consistent with the $S(T)$ curves. Most importantly, when Bi$_{1-x}$Sb$_x$ is incorporated as a secondary phase, $\kappa(T)$ decreases significantly, which we attribute to the complex microstructural evolution involving microscale Bi$_{1-x}$Sb$_x$ GBs with an extremely large acoustic mismatch (≈9 THz) with respect to the Heusler matrix, periodic composition fluctuations within the grains, and an enhanced solubility limit of heavy Ta atoms. Moreover, $\kappa(T)$ of the Bi$_{1-x}$Sb$_x$ GB network is likely reduced as well compared to the bulk values owing to the ladder-like arrays of stacking fault defects (see Fig. 2f, g) and microscale composition fluctuations (Fig. S9), likely inhibiting phonon-driven heat transport along the GBs[36].

From Fig. 4b, it is evident that, despite the significant reduction in $\kappa(T)$, electronic transport remains excellent. The temperature-dependent resistivity $\rho(T)$ flattens when Bi$_{0.9}$Sb$_{0.1}$ is incorporated, even resulting in a decrease of $\rho(T)$ at elevated temperatures. The flattening of the resistivity curves and the increased residual resistivity at low temperatures both imply a weakening of the electron-phonon coupling and enhanced disorder, aligning with the notion of an enhanced Ta solubility limit. Previous work shows that V/Ta substitution results in a softening of the Heusler lattice and a reduction of the

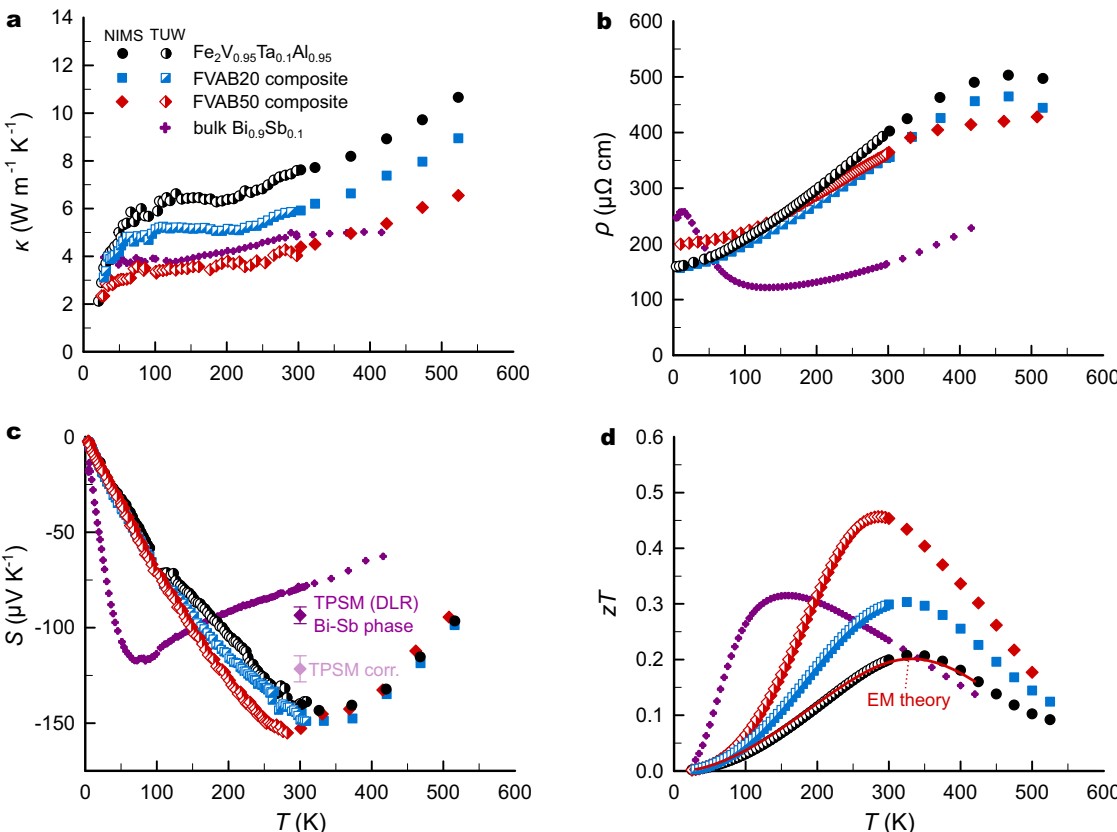

**Fig. 4 | Thermoelectric properties of ($Fe_2V_{0.95}Ta_{0.1}Al_{0.95}$ + $Bi_{0.9}Sb_{0.1}$) composites. a** Temperature-dependent thermal conductivity, **b** electrical resistivity, **c** Seebeck coefficient, and (**d**) dimensionless figure of merit $zT$ of $Fe_2V_{0.95}Ta_{0.1}Al_{0.95}$ composites with 20 and 50 vol.% $Bi_{0.9}Sb_{0.1}$ added before sintering (FVAB20, FVAB50) compared to pristine $Fe_2V_{0.95}Ta_{0.1}Al_{0.95}$ and $Bi_{0.9}Sb_{0.1}$. The error bars in (**c**) indicate the statistical variation of the Bi-Sb phase corresponding to the area shown in Fig. 3b. The red solid line in (**d**) represents a calculation based on effective-medium theory (EMT) for FVAB50, using a volume fraction of ≈6 vol.% $Bi_{0.9}Sb_{0.1}$, determined by EDX.

electron-phonon deformation potential, decreasing $\rho(T)$ at elevated temperatures, where acoustic phonon scattering dominates. Moreover, V/Ta substitution can expand the band gap by pushing the V-$e_g$ conduction band toward higher energies, enhancing the maximum of the Seebeck coefficient[37].

The temperature-dependent Seebeck coefficient $S(T)$ only varies moderately in the composites with $S_{max}$ shifting to slightly lower temperatures. As mentioned in the previous section, a simple effective-medium theory (EMT) with parallel conduction along the GB network would result in a sizeable decrease of $S(T)$. The fact that $S$ retains large values, is surprising and unexpected, calling for more advanced theoretical studies of TE transport in composite materials. Although it is well known that the TE properties of nanocomposites can deviate strongly from those of the individual material components[38,39], deviations from the EMT in microscale composites are much rarer.

The above-listed modifications result in an extreme boost of $zT$ by more than a factor of two (see Fig. 4d) up to a $zT_{max}$ of almost 0.5 at 295 K. Note that, like for almost all TE studies, error bars of ≈20% should be considered[40], which were omitted for better visibility of the data. This clearly exceeds the predictions of the EMT, which, as demonstrated by Bergman and Levy in their seminal work[41], states that $zT$ in composites needs to be always smaller than the largest $zT$ of the individual components, irrespective of the geometry. We also note that this exceeds the largest room-temperature $zT$ of the entire binary Bi-Sb system, with $zT_{max} \approx 0.3$[33]. This implies that the TE properties of the individual components change dramatically in the composite or are subject to reciprocal action of both, allowing for a decoupling of charge and heat transport. To investigate the thermal stability of our

composites, transport measurements were conducted for various thermal cycles (Fig. S22), which reveal excellent reproducibility and no degradation of the properties at least up to 500 K – the most relevant temperature range for the potential application of these materials.

## Field-dependent magneto-transport

To further elucidate transport in the best-performing composite sample (FVAB50), we measured the Hall effect in a broad temperature and magnetic field range, 4–400 K and −9 T to 9 T. These results are summarized in Fig. 5. The field-dependent Hall resistivity $\rho_{xy}$, plotted in Fig. 5a for various temperatures displays an extremely large anomalous Hall effect, which even increases with rising temperature up to ≈300 K, despite the absence of any sizeable magnetization in the sample. On the contrary, the sintered Heusler compound without $Bi_{1-x}Sb_x$ at the GBs exhibits a simple linear magnetic field dependence. While the observation of a giant anomalous Hall effect in various topological materials is often ascribed to huge Berry curvatures, emerging from the respective topological band structure features[42,43], we interpret the complex field-dependent curves in Fig. 5a as a two-channel conduction mechanism, where charge carriers can move across the sample either through topologically trivial bulk states of the Heusler grains or through topologically protected surface states of the $Bi_{1-x}Sb_x$ GB network (see inset Fig. 5b). Figure 5b shows that the field-dependent behavior of $\rho_{xy}$ from −5 T to 5 T can be reasonably well described by a simple two-channel transport model (details of the modeling procedure and underlying theory is presented in the SI). The mobilities obtained for the two distinct transport channels are presented in Fig. 5c alongside the values of pristine $Fe_2V_{0.95}Ta_{0.1}Al_{0.95}$ without topological-insulating GBs. The bulk values of $Fe_2V_{0.95}Ta_{0.1}Al_{0.95}$ are of

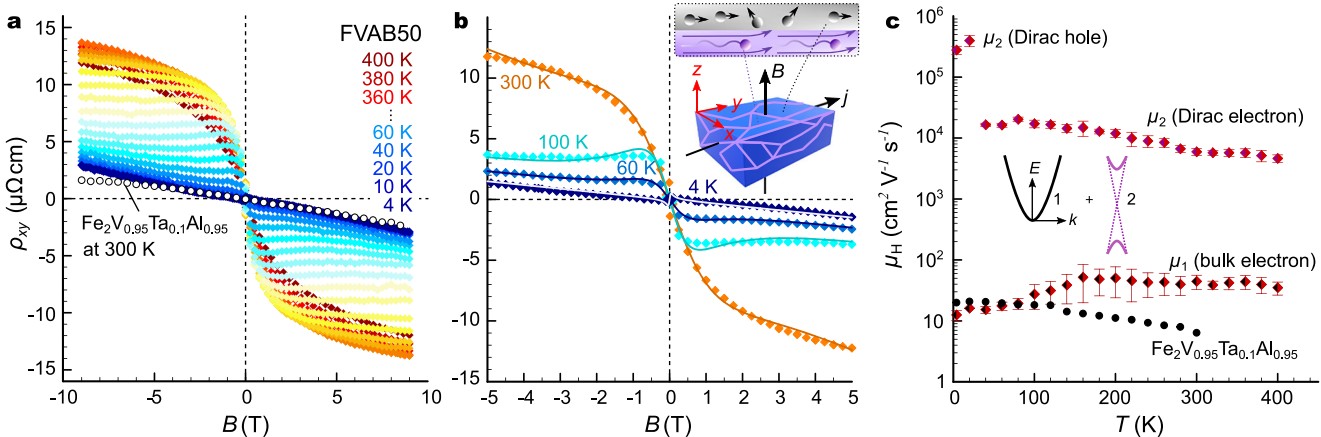

**Fig. 5 | Magneto-transport properties of FVAB50 composite. a** Field-dependent Hall resistivity of FVAB50 at different temperatures $4 \leq T \leq 400$ K. Room-temperature data of $Fe_2V_{0.95}Ta_{0.1}Al_{0.95}$ are shown for comparison as black open circles. **b** Two-channel transport modeling of field-dependent Hall resistivity. Solid lines are least squares fits. Inset shows a sketch of the Hall effect in FVAB$X$ composites. Highly mobile Dirac-like carriers along the Bi-Sb GB network have a much larger mean free path than the Heusler bulk electrons and are deflected much easier, even in small magnetic fields. **c** Temperature-dependent Hall mobility of FVAB50 obtained by modeling the complex field-dependent behavior. Error bars indicate uncertainties in the fit results. The inset shows a sketch of the two transport channels (1) the bulk conduction electrons of the Heusler main phase and (2) the Dirac-like surface states of the Bi-Sb GB network.

the order of $10$ $cm^2$ $V^{-1}$ $s^{-1}$. The mobility of the bulk channel, $\mu_1$, extracted from our transport modeling is comparable, especially at low temperatures. The mobility of the Dirac-like surface states, $\mu_2$, associated with the Bi-Sb network, on the other hand, is several orders of magnitude higher up to $3 \times 10^5$ $cm^2$ $V^{-1}$ $s^{-1}$ and $2 \times 10^4$ $cm^2$ $V^{-1}$ $s^{-1}$ for the Dirac holes and electrons, respectively. As a consistency check, we calculated the temperature-dependent zero-field resistivity from the obtained carrier mobilities $\mu_{1,2}$ and carrier densities $n_{1,2}$ via $\rho_{xx}(0, T) = (en_1\mu_1 + en_2\mu_2)^{-1}$, which should match temperature-dependent measurements in Fig. 4b. As shown in Fig. S19, there is excellent agreement across the entire temperature range, underscoring the robustness and reliability of the fits.

In summary, the field-dependent Hall effect reveals a significant contribution to electronic transport from the Dirac-like surface states of the $Bi_{1-x}Sb_x$ GB network, leading to a pronounced anomalous Hall effect, which can be explained by a two-channel transport model. This aligns with the colossal mobilities expected from such topologically robust carriers and the higher surface-area-to-volume ratio in the composite.

## Discussion

Concluding, we demonstrated that incorporating $Bi_{1-x}Sb_x$ in $Fe_2VAl$ Heusler compounds can boost the $zT$ compared to both individual materials. This is particularly surprising considering the near-complete immiscibility of both components and their chemical distinctness, which should prevent sizeable interdiffusion and changes to the individual material properties. Decoupling charge and lattice-driven heat transport in such composites is an auspicious route toward high $zT$, even more so in systems where reducing grain size and alloying strongly compromise carrier mobility, although, a more profound theoretical understanding of charge and heat transport in composites is required to optimally design and choose the best candidates.

In this study, we achieved heavily reduced $\kappa_L$, and simultaneously high $\mu_W$. To provide a broad comparison with other material classes for the latter, we downloaded all available TE property data from the Starrydata2 open web database[44], which, as of July 2024, contains TE data from 8961 different papers and 52,020 different samples. We then calculated $\mu_W$ for those samples, where both $S(T)$ and $\rho(T)$ are reported. Fig. S23 shows that, near room temperature, $\mu_W$ of the best composite sample from this work surpasses all other reported $n$-type semiconductors.

To further elevate the performance of $Fe_2VAl$ systems, broad screening of secondary phases needs to be done; especially other topological insulators like $Bi_2Se_3$ could be considered. Additionally, one has to think about strategies to increase and reliably tune the volume fraction. Lastly, it is crucial to identify promising $p$-type compounds with competitive $z$. Since $p$-type $Fe_2VAl$ compounds inherently show much smaller Seebeck coefficients, this can only be realized via band engineering of the valence band electronic structure[45,46]. Only then can competitive $Fe_2VAl$-based modules be realized, which could substitute the long-reigning $Bi_2Te_3$ systems. The present study suggests that, by proper GB engineering, $Fe_2VAl$ Heusler alloys may indeed bear the potential to rival state-of-the-art $Bi_2Te_3$ and $Mg_3Bi_2$ semiconductors. High-performance modules entirely based upon $Fe_2VAl$ alloys could open a new era for near-ambient applications, as these systems excel in terms of cost-effectiveness, excellent recyclability, and simpler device structures. Moreover, they exhibit superior mechanical, thermal, and chemical long-term stability, factors that are becoming increasingly recognized as essential assets for realizing widespread thermoelectric technology.

## Methods

### Synthesis of starting materials and composites

Bulk elements of high purity (Fe 99.99%, V 99.93%, Ta 99.95%, Al 99.999%, Bi 99.999%, Sb 99.999%) were stoichiometrically weighed and polycrystalline ingots of the starting materials ($Fe_2V_{0.95}Ta_{0.1}Al_{0.95}$ and $Bi_{0.9}Sb_{0.1}$) were synthesized by high-frequency induction melting under Ar atmosphere. The as-cast ingots were manually crushed and ground using a tungsten carbide pestle and mortar. The resulting powders from the individual starting materials were then mixed in various volume ratios. The volume percentage of $Bi_{0.9}Sb_{0.1}$ powder added was adjusted and calculated based on the theoretical densities of the respective starting materials. After the powders were thoroughly mixed, the mixture was filled into a graphite die and sintered at a temperature of 1373 K, which is about 80% of the melting point of the full-Heusler phase and about 800 K higher than the melting point of $Bi_{0.9}Sb_{0.1}$. Consequently, excess liquid was expelled during the sintering process. Additionally, we observed that the liquid-phase sintering led to a significant decrease in the sintering temperature of the Heusler material by up to almost 200 K when $Bi_{0.9}Sb_{0.1}$ powder was added as compared to when only $Fe_2V_{0.95}Ta_{0.1}Al_{0.95}$ powder was sintered. Nonetheless, to ensure consistent and comparable processing

conditions for the samples studied in this work, all specimens were sintered using exactly the same synthesis conditions, i.e., a compaction pressure of 50 MPa, a maximum temperature of 1373 K, and a holding time of 15 min. No additional heat treatment has been applied to the samples after the sintering process.

## Structural characterization

The microstructure and elemental composition of the sintered samples were investigated using scanning electron microscopy in both secondary electron (SE) and backscattered electron (BSE) modes, complemented by energy-dispersive X-ray spectroscopy (EDX). These analyses were performed on an ultra-high-resolution field emission SEM (HRSEM SU8230, Hitachi, Japan), equipped with an X-Max$^N$ EDS detector (Horiba, Japan). For HRSEM observations, the samples were mounted in electroconductive epoxy and polished meticulously. EDX analysis utilized an acceleration voltage of 25 kV, gathering $10 \times 10^6$ counts per EDX map and $1 \times 10^6$ counts for point analysis.

To investigate the interface between the $Fe_2VAl$ matrix and the $Bi_{0.9}Sb_{0.1}$ secondary phase at the nanoscale, the sample was prepared using a conventional focused ion beam (FIB) technique. A thin section was extracted from the targeted area, attached to an Omnigrid, and thinned to approximately 90 nm for electron transparency. Additionally, the FVAB50 sample was crushed into fine particles, dispersed in ethanol, and deposited on a grid to investigate the sample surface. Transmission electron microscopy (TEM) bright-field and lattice images were acquired using a JEOL JEM-3100FEF (JEOL, Japan) microscope operating at 300 kV, which was also equipped with an EDS detector for detailed elemental mapping.

The X-ray powder diffraction measurements were conducted at the Institute of Solid State Physics, TU Wien, using an in-house diffractometer (AERIS by PANalytical). These measurements utilized standard Cu K-$\alpha$ radiation, with data collected in the Bragg-Brentano geometry over the angular range $20° < 2\theta < 100°$. Rietveld refinements on the obtained powder patterns were performed using the program PowderCell.

## High-temperature property measurements

Thermal conductivity measurements at high temperatures were performed in $N_2$ atmosphere directly on the sintered pellets, in the direction parallel to the pressing (compaction) direction during sintering with a commercially available setup (LFA 467 by NETZSCH). The instrument makes use of a conventional laser flash method for the diffusivity $D$ and a differential scanning calorimeter for determining the specific heat $c_p$. The density of the sample $d_m$ was determined via Archimedes principle and the thermal conductivity was calculated from $\kappa = D c_p d_m$.

After performing high-temperature thermal conductivity measurements, the samples were cut into strips 2–3 mm in width and 8–10 mm in length using a high-speed aluminum oxide cutting wheel. The bar-shaped samples were then mounted in a commercial setup (ZEM3 by ADVANCE RIKO) and the electrical resistivity and Seebeck coefficient were measured as a function of temperature. For the best sample, the measurement was repeated to confirm reproducible and stable results.

## Low-temperature property measurements

Low-temperature measurements provide valuable insights into lower-energy excited states and states near the Fermi energy. This is especially significant for samples with narrow energy gaps, such as $Fe_2VAl$-based full-Heusler and binary $Bi_{1-x}Sb_x$ systems, where the Seebeck coefficient often peaks below or near room temperature. Furthermore, when modeling temperature-dependent data (using a parabolic band model for example), it is crucial that the experimental data span a wide temperature range. The thermoelectric characterization at low

temperatures was carried out at TU Wien (Austria) on the same rectangular bar-shaped sample pieces that were used for the high-temperature measurements at NIMS (Japan).

The temperature-dependent electrical resistivity was measured in a home-built bath cryostat at TU Wien. The sample was contacted in a four-probe geometry with thin gold wires, using a spot-welding device. The sample was then mounted on a sample puck using GE Varnish as an adhesive and directly inserted into the cryostat. The measurement was performed continuously every time the temperature changed by 1 K.

The temperature-dependent Seebeck coefficient was also measured on the very same sample piece using a different home-built setup at TU Wien. Here, two chromel-constantan thermocouples are contacted to both ends of the sample to pick up the temperature difference and voltages. Since it is difficult to solder directly on the sample surface, a bundle of thick copper wires was first spot-welded onto the sample surface to which the thermocouples were then soldered. The high thermal conductivity of copper and the fact that the thermocouples are soldered in very close proximity to the sample surface ensures that the cold finger effect can be minimized. Furthermore, two strain gauges with a resistance of $\approx 120\,\Omega$ function as heaters and are fixed to the bottom of both sample ends via GE varnish. The two heaters allow switching the temperature difference ("seesaw heating"[47]) to cancel spurious voltage contributions. The measurement is carried out in an evacuated sample chamber with He exchange gas to ensure thermal coupling to the cryogen.

The thermal conductivity at low temperatures was measured by making use of a steady-state method using a home-built sample probe with a flow cryostat. Here, a heater is attached to the top surface of the sample employing a thermally conductive epoxy resin (STYCAST 2850FT). Similar to the Seebeck coefficient measurements, two bundles of copper wires are first fixed to the sample to each of which a thermocouple is then soldered. The bottom of the sample is mounted on a copper heat sink and the measurement is carried out in high vacuum ( $\approx 10^{-5}$ mbar).

## Modeling of field-dependent Hall resistivity

In a two-channel transport model for two types of charge carriers with charge $q_{1,2}$, carrier density $n_{1,2}$, and carrier mobility $\mu_{1,2}$, the field-dependent longitudinal resistivity $\rho_{xx}(B)$ and Hall resistivity $\rho_{xy}(B)$ can be expressed as[48]

$$\rho_{xx}(B) = \frac{q_1 n_1 \mu_1 + q_2 n_2 \mu_2 + (q_1 n_1 \mu_2 + q_2 n_2 \mu_1)\mu_1 \mu_2 B^2}{(q_1 n_1 \mu_1 + q_2 n_2 \mu_2)^2 + (q_1 n_1 + q_2 n_2)^2 \mu_1^2 \mu_2^2 B^2} \quad (1)$$

and

$$\rho_{xy}(B) = \frac{q_1 n_1 \mu_1^2 + q_2 n_2 \mu_2^2 + (q_1 n_1 + q_2 n_2)^2 \mu_1^2 \mu_2^2 B^2}{(q_1 n_1 \mu_1 + q_2 n_2 \mu_2)^2 + (q_1 n_1 + q_2 n_2)^2 \mu_1^2 \mu_2^2 B^2} B . \quad (2)$$

From the above equations, it becomes evident that non-linearities can occur in the field-dependent Hall resistivity if, for instance, there is a sizeable difference between $\mu_1$ and $\mu_2$. While previous models have employed four different fit parameters, i.e. the respective carrier densities and mobilities to model non-linear field dependencies, Eguchi and Paschen previously suggested a novel, more robust scheme for analyzing field-dependent magneto-transport properties[49]. We utilized this framework to model the non-linear field-dependent Hall effect of the FVAB50 composite. More details regarding the modeling procedure are presented in the Supplementary Information.

## Data availability

All data supporting the findings of this study are available within the article and its Supplementary Information file.

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

## Acknowledgements

Research in this paper was financially supported by the Japan Science and Technology Agency (JST) program MIRAI, JPMJMI19A1. Furthermore, F.G. acknowledges financial support by the Lions Club Wien St. Stephan and J.d.B. acknowledges funding by the Deutsche Forschungsgemeinschaft (DFG, German Research Foundation), project number 520487260 The authors also acknowledge the TU Wien Bibliothek for financial support through its Open Access Funding Programme.

## Author contributions

F.G., M.P., and A.R. conceived the idea for the study. F.G. designed the work, and, with help from I.S., C.B., and Y.H., synthesized the samples and measured the thermoelectric properties. I.S., S.G., and H.D.N. investigated the micro- and nanostructure of the material via SEM and TEM techniques. S.G., P.Z., and G.O. performed TPSM measurements, and E.M. and J.d.B. analyzed and interpreted the data. C.B., S.S., G.R., and E.S. assisted in the synthesis and experimental investigations of the samples. P.R., N.K., E.M., and E.B. discussed and improved the contents of the paper. J.d.B. and T.M. supervised the work and assisted in outlining the initial draft of the manuscript. F.G. wrote the initial draft. All authors read, discussed, and edited the manuscript.

## Competing interests

The authors declare no competing interests.
