## [Transparent Peer Review file · Nature Communications]

Decoupled charge and heat transport in Fe₂VAl composite thermoelectrics with topological-insulating grain boundary networks

Corresponding Author: Dr Fabian Garmroudi

Version 0:

Reviewer comments:

Reviewer #1

(Remarks to the Author)

Review report for Manuscript: "Topological-insulating grain boundary networks for high-performance Fe₂VAl thermoelectrics", by Fabian and Mori et al., submitted for review purpose in Nature Communications.

The manuscript demonstrates a reduction in lattice thermal conductivity (κ_L) alongside an unexpected increase in carrier mobility (μ_W) by incorporating chemically and structurally distinct Bi_{1-x}Sb_x at the grain boundaries. This approach effectively decouples charge and heat transport, enhancing thermoelectric performance. The authors report a figure of merit of $zT \approx 0.5$ at room temperature for n-type half- and full-Heusler compounds. The incorporation of Bi_{1-x}Sb_x, a known topological insulator, into the Heusler compound Fe₂VAl is referred to as "liquid-phase engineering" due to its lower melting point compared to the Heusler compound. The analysis is based on a comprehensive investigation, including structural characterization, thermoelectric and magneto-transport measurements, which reveal significant modifications in the microstructure. These changes include a reduction in lattice thermal conductivity and a concurrent increase in carrier mobility, attributed to topologically protected charge transport along grain boundaries. This thorough study encompasses material preparation, electrical and thermal transport measurements, and modeling to substantiate the hypothesis regarding the interfacial contributions to the thermoelectric performance of the materials. The work is compelling and can be considered after addressing the following clarifications.

The comments are listed below:

1. The title has topological word and misleading. In the present work, nothing explicitly studied for the topological properties of Bi_{1-x}Sb_x but an effect of inclusion at the grain boundaries.
2. Authors have implied the incorporation of Bi_{1-x}Sb_x topological insulator in the grain boundary network of the parent compound, however they have not stated why only the Bi_{1-x}Sb_x is going at grain boundaries and is not remaining as a precipitate or condensate in clusters form in the compound. Is there any specific reason for that.
3. From the EDS maps, Ta is also going in the grain boundary network. Does this have any contribution to the enhanced performance and reduced thermal conductivity? Thus, author may like to comment on the contribution of Ta (being heavier element) in the overall performance.
4. In fig 4 d the maximum zT is ~ 0.45 , but the authors have mentioned ~ 0.5 in the abstract. The figure 4 can be shown with the tentative errors in the measurements. Figure 1 B, the z has maxima at ~ 240 K while the ZT graph in figure 4-D has maxima at ~ 290 K. This requires a careful examination of the data and revisions in the narratives.
5. Abbreviation can be defined when used first time.
6. Authors need to mention the zT of undoped Fe₂VAl for comparison. They have used both 'Z' and 'ZT', however for convention and comparison, it is good to mention ZT only.

7. The symbol size for bulk Bi_{0.9}Sb_{0.1} can be increased in Fig 4.

8. Figure 4 B, the authors have mentioned that the resistivity is decreasing above 300 K, with the incorporation of Bi_{0.9}Sb_{0.1}. This require clarifications. What could be the possible reason behind such a trend?

9. In the Supplementary Information, authors performed Rietveld refinement for pure and 50% samples and showed variation in the lattice parameters. The authors may like to try to estimate the presence of impurities by adding the secondary phase of BiSb in the refinement process. Similarly, this can be tested for the 20% sample to get the exact trend in variation in the lattice parameters. Additionally, this will also provide the estimation of tentative percentage of secondary phase present in the parent Heusler compound.

10. The interfaces are anomalous and thermodynamically unstable [Nano Letters 12 (8) 4305 (2012)], which affects the overall performance on multiple cycling of the operations. Authors may like to address the interfacial instabilities during operations.

11. Authors can also add the designation of symbols used for various compounds in Fig 1 a.

Reviewer #2

(Remarks to the Author)

Manuscript Number: NCOMMS-24-72976-T

Title: Topological-insulating grain boundary networks for high-performance Fe₂VAl thermoelectrics

The study suggests the enhanced carrier mobility by the topologically protected charge transport in the topological insulator Bi_{1-x}Sb_x between Fe₂V_{0.95}Ta_{0.1}Al_{0.95} grains. The topologically protected charge transport is an interesting and important concept for the development of high-performance thermoelectric materials. However, the manuscript dose not include reasonable measurement results, making the findings difficult to accept. Therefore, the manuscript is not recommended for publication in this journal due to the following issues.

1) Firstly, the sample was fabricated using liquid-phase sintering with Fe₂V_{0.95}Ta_{0.1}Al_{0.95} powder and Bi_{0.9}Sb_{0.1}. The density of the sample should be provided for sintered samples. Additionally, since the composite consists of different materials, the material ratio can be inferred from SEM/EDX results. The composite material ratio is a key factor in understanding the transport properties of the composite samples.

2) High-performance thermoelectric materials often exhibit anisotropic behavior in sintered samples due to the preferred orientation of their layered structure along the pressing direction. Since Bi_{0.9}Sb_{0.1} has a layered structure, the thermoelectric properties should be measured in the same direction to account for the anisotropic behavior of the sintered sample. The authors used a sintered pellet for the LFA measurement and a sample cut from the LFA specimen, meaning that the measurement directions for the ZEM-3 and LFA are perpendicular. Additionally, samples prepared via liquid-phase sintering could be inhomogeneous [Science 348 (2015) 109-114]. To ensure reliable data for the sintered composite, measurements at different sample positions and re-measurements are necessary to confirm homogeneity and thermal stability.

3) The lattice parameters of the FVAB50 composite and Fe₂V_{0.95}Ta_{0.1}Al_{0.95} differ in Fig. 2(e). This suggests that the thermoelectric properties of the composite may be influenced by doping during the sintering process. Consequently, the enhanced carrier mobility might result from the reduced carrier concentration in the composite.

4) The composition of the Bi_{0.9}Sb_{0.1} may change during the melting process. The electrical properties of the Bi_{1-x}Sb_x are significantly influenced by the composition (x) [Journal of Alloys and Compounds 467 (2009) 305–309]. Therefore, since the composition ratio of the Bi_{1-x}Sb_x is a crucial factor in this study, the chemical composition should be provided.

Reviewer #3

(Remarks to the Author)

In this manuscript, a new approach was proposed to decouple the charge and heat transport of Fe₂VAl full-Heusler compounds by incorporating archetypal topological insulator Bi_{1-x}Sb_x. Although the underlying mechanism of the synergistic optimization of transport properties is still unclear, the presented work is interesting and makes a positive contribution to the field of thermoelectricity, in particular the realization of an exciting room-temperature zT for Fe₂VAl alloy. However, before I recommend the publication to Nature Communications, there are some concerns as listed below:

1. The effective-medium theory states that zT in composites are always smaller than the largest zT of the individual components. But actually this is not suitable for all many composites, especially for the nanocomposites, for example, BaFe₁₂O₁₉/Ba_{0.3}In_{0.3}Co₄Sb₁₂ in Nat Nanotechnol 2017, 12, 55 and (Fe,Co,Ni)/Ba_{0.3}In_{0.3}Co₄Sb₁₂ in Nature 2017, 549, 247. zT values for the nanocomposites are apparently higher than those of the individual components. Thus, the exceeding the predictions of effective-medium theory, as claimed by authors should be a common phenomenon for composite materials.

2. Long-term stability and reliability are crucial for the practical application of thermoelectric materials. The assessment of the stability and reliability of the composite material over extended periods is lacking. I suggest that the authors supplement the manuscript with long-term thermoelectric performance data, including but not limited to changes in microstructure thermal

conductivity, electrical conductivity, and Seebeck coefficient over time.

3. The author pointed out that by incorporating archetypal topological insulator $\text{Bi}_{1-x}\text{Sb}_x$ between $\text{Fe}_2\text{V}_{1-x}\text{Ta}_x\text{Al}$ grains, the solid solubility of Ta is also increased. However, it remains to be further proved whether the enhanced thermoelectric performance is dominated by the two-channel transport model of the composite material as proposed in manuscript, or the enhanced doping effect caused by Ta. So, the influence and contribution of Ta doping to the thermoelectric properties should be detailed discussed.

4. The paper utilizes the two-band model to calculate the temperature-dependent Seebeck coefficient for $\text{Fe}_2\text{V}_{0.95}\text{Ta}_{0.1}\text{Al}_{0.95}$ and $\text{Bi}_{0.9}\text{Sb}_{0.1}$, with experimental values being consistent with theoretical values. However, the band structure information for these materials is missing. I suggest that the authors provide the band structure information for these two materials. Furthermore, Fig. S9 indicates a significant fluctuation in the ratio of Bi to Sb at the grain boundaries. At this point, the comparison of the Seebeck coefficient calculated using the two-band model for $\text{Bi}_{0.9}\text{Sb}_{0.1}$ with experimental values seems to be no reliability.

Version 1:

Reviewer comments:

Reviewer #1

(Remarks to the Author)

The authors have incorporated the reviewers' suggestions and addressed their concerns satisfactorily. The revised version is now suitable for consideration.

Reviewer #2

(Remarks to the Author)

In my opinion, the manuscript has been improved and is acceptable after the corrections. Therefore, I recommend its publication in Nature Communications.

Response to Review (NCOMMS-24-72976-T)

Reviewer #1 (Remarks to the Author):

The manuscript demonstrates a reduction in lattice thermal conductivity (κ_L) alongside an unexpected increase in carrier mobility (μ_w) by incorporating chemically and structurally distinct $\text{Bi}_{1-x}\text{Sb}_x$ at the grain boundaries. This approach effectively decouples charge and heat transport, enhancing thermoelectric performance. The authors report a figure of merit of $zT \approx 0.5$ at room temperature for n-type half- and full-Heusler compounds. The incorporation of $\text{Bi}_{1-x}\text{Sb}_x$, a known topological insulator, into the Heusler compound Fe_2VAl is referred to as "liquid-phase engineering" due to its lower melting point compared to the Heusler compound. The analysis is based on a comprehensive investigation, including structural characterization, thermoelectric and magneto-transport measurements, which reveal significant modifications in the microstructure. These changes include a reduction in lattice thermal conductivity and a concurrent increase in carrier mobility, attributed to topologically protected charge transport along grain boundaries. This thorough study encompasses material preparation, electrical and thermal transport measurements, and modeling to substantiate the hypothesis regarding the interfacial contributions to the thermoelectric performance of the materials. The work is compelling and can be considered after addressing the following clarifications.

Reply: We sincerely thank the reviewer for the thoughtful and detailed evaluation of our manuscript. Below, we provide our detailed responses to each of the reviewer's comments and clarifications.

The comments are listed below:

1. The title has topological word and misleading. In the present work, nothing explicitly studied for the topological properties of $\text{Bi}_{1-x}\text{Sb}_x$ but an effect of inclusion at the grain boundaries.

Reply: We agree that a different title could be more suitable and have been pondering about a better one. How about: "*Decoupled charge and heat transport in Fe_2VAl composite thermoelectrics with topological-insulating grain boundary networks*", which is more informative and correct. We welcome any additional title suggestions from the reviewer. Regarding the topological properties of the $\text{Bi}_{1-x}\text{Sb}_x$ grain boundary phase, our idea is that the topological-insulating nature of $\text{Bi}_{1-x}\text{Sb}_x$ plays a more crucial role in the composite, where the surface to volume ratio is greater. Despite a strong reduction of the lattice thermal conductivity, electronic transport remains excellent, and the weighted mobility is even enhanced. Studying the field-dependent Hall effect in a broad temperature and field range, we observe a highly anomalous response, which we interpret as a two-channel transport,

from which ultrahigh carrier mobilities are deduced, suggesting that charge transport along the grain boundaries takes place in a topologically protected manner. However, we acknowledge that the hypothesis that this is elevated significantly in the composite compared to the bulk requires further investigation, and we hope our work can inspire future studies of composites incorporating topological insulators at grain boundaries.

2. Authors have implied the incorporation of $\text{Bi}_{1-x}\text{Sb}_x$ topological insulator in the grain boundary network of the parent compound, however they have not stated why only the $\text{Bi}_{1-x}\text{Sb}_x$ is going at grain boundaries and is not remaining as a precipitate or condensate in clusters form in the compound. Is there any specific reason for that.

Reply: It is not possible that the $\text{Bi}_{1-x}\text{Sb}_x$ phase forms precipitates within the Heusler grains since the particles of the Heusler phase do not melt during sintering, and these two materials are nearly completely immiscible, preventing sizeable interdiffusion, as confirmed from HR-TEM investigations.

To address the question, why the $\text{Bi}_{1-x}\text{Sb}_x$ phase does not form clusters, we have compared the microstructures of various composite samples with different amounts of $\text{Bi}_{0.9}\text{Sb}_{0.1}$ powder added before the sintering process (Fig. S10 in the Supplementary Information). From these analyses it becomes clear that, *initially*, the liquid $\text{Bi}_{1-x}\text{Sb}_x$ phase, fills primarily the triple junctions between the grains during sintering and indeed forms clusters. However, interestingly, as the $\text{Bi}_{1-x}\text{Sb}_x$ amount increases, the liquid phase coats the grains and forms a grain boundary network. Thus, one may conclude that the wettability of the Heusler phase changes with the ratio between the solid Heusler particles and the liquid $\text{Bi}_{1-x}\text{Sb}_x$ phase.

3. From the EDS maps, Ta is also going in the grain boundary network. Does this have any contribution to the enhanced performance and reduced thermal conductivity? Thus, author may like to comment on the contribution of Ta (being heavier element) in the overall performance.

Reply: The samples are very close to the solubility limit of Ta in the Heusler structure. Thus, small precipitates of a Ta-enriched Fe_2Ta -type Laves phase form. By extensive SEM investigations, we noticed that the formation of these precipitates becomes suppressed (although not completely) upon adding $\text{Bi}_{0.9}\text{Sb}_{0.1}$ during the sintering process. We suggest that this can be traced back to the fact that the grain boundaries, where segregation of the Ta-rich phase is energetically more favorable, become occupied with the $\text{Bi}_{1-x}\text{Sb}_x$ phase, which suppresses precipitation of the Ta-rich Laves phase. This is confirmed by a clearly reduced number of such precipitates at the grain boundaries as well as by supersaturated regions within the Heusler grains, that show a higher concentration of Ta in the EDX line scans (Fig. 2d in the manuscript). The supersaturation of the Heusler structure with the

larger and heavier Ta atoms indeed contributes to the reduction of the lattice thermal conductivity and enhances thermoelectric performance. As suggested by the referee, we have stressed this a little more in the revision by adding appropriate adjustments to the text: bottom right of page 3 “Both these structural changes suggest an enhanced solubility limit of **heavy Ta atoms**, when $\text{Bi}_{1-x}\text{Sb}_x$ is incorporated as a GB network during the liquid-phase sintering, contributing to a reduction of the lattice thermal conductivity as shown later”, top left of page 4 “as larger **and heavier Ta atoms** are substituted” and bottom left of page 5 “and an enhanced solubility limit of **heavy Ta atoms**”. Regarding the role of the precipitates themselves, we are currently planning detailed studies to study thermal transport at the interface between the Heusler phase and the Ta-rich precipitates to examine whether there is a large thermal boundary resistance. To this aim, a new in situ STEM technique for studying thermal wave behavior will be employed, which has recently been developed by some of the co-authors [Science Advances 10, eadj3825 (2024)].

4. In fig 4 d the maximum zT is ~ 0.45 , but the authors have mentioned ~ 0.5 in the abstract. The figure 4 can be shown with the tentative errors in the measurements. Figure 1 B, the z has maxima at ~ 240 K while the ZT graph in figure 4-D has maxima at ~ 290 K. This requires a careful examination of the data and revisions in the narratives.

Reply: We appreciate the reviewer’s careful observation that this might have been a bit confusing. We have tried to make this clearer in the updated version. To be precise, the maximum zT is around 0.46 between 280 and 300 K, whereas the maximum z value is 1.72 between 240 and 250 K (due to the missing linear T factor the maximum is shifted towards lower temperatures). It is well known that one must consider error bars for the figure of merit of at least $\sim 20\%$ because its calculation involves multiple different physical properties [Energy & Environmental Science 8, 423-435 (2015)], and we have rounded all values to the first digit. Similar error bars apply to all thermoelectric studies, and they are commonly omitted by the community (who is well aware of these error bars). Since we reckon that error bars add little value to the figures, maybe even obscuring them and making them less appealing to look at, we have omitted them purposefully in this study as well. The following statement has been added in the manuscript (top left, page 6) “Note that, like for almost all TE studies, error bars of $\sim 20\%$ should be considered [Energy & Environmental Science 8, 423-435 (2015)], which were omitted for better visibility of the data.”

5. Abbreviation can be defined when used first time.

Reply: We have reviewed the manuscript and confirmed that all abbreviations are defined upon first use. If the reviewer has specific abbreviations in mind, we would be happy to make further adjustments.

6. Authors need to mention the zT of undoped Fe_2VAl for comparison. They have used both 'Z' and 'ZT', however for convention and comparison, it is good to mention ZT only.

Reply: Thank you for the suggestion. We have contemplated adding a short discussion of the zT of undoped Fe_2VAl in the text. The zT of undoped Fe_2VAl is negligibly small (of the order of $10^{-3} - 10^{-2}$) due to its high intrinsic lattice thermal conductivity and suboptimal Fermi level position. However, it seems unreasonable to boast about a huge improvement compared to these negligibly small values. In our humble opinion, it is unreasonable to compare samples which are close to the optimal carrier concentration with those that are not. For this reason, we have included in Fig. 1b, only samples that are doped close to the optimal carrier concentration. To clarify this, we have changed “*compared to other high-performance n-type thermoelectrics*” to “*compared to other **optimally doped**, high-performance n-type thermoelectrics*” in the caption of Fig. 1b.

Regarding the use of “ z ” and “ zT ”, we intentionally used z in Fig. 1 as it allows a clearer comparison of different material classes (in a visual plot) with different temperatures of optimal performance. The tradeoff between lattice thermal conductivity and weighted mobility is also more directly reflected in z , whereas the T factor in zT is merely added to obtain a dimensionless quantity. We prefer showing z in Fig. 1 and near the beginning of our manuscript, where the focus lies on the tradeoff between lattice-driven heat and charge transport, particularly since a comparison of zT of our samples is shown and discussed anyways later in the manuscript.

7. The symbol size for bulk $\text{Bi}_{0.9}\text{Sb}_{0.1}$ can be increased in Fig 4.

Reply: Thanks for your note. We have increased the size of the symbols for better visibility, as requested.

8. Figure 4 B, the authors have mentioned that the resistivity is decreasing above 300 K, with the incorporation of $\text{Bi}_{0.9}\text{Sb}_{0.1}$. This require clarifications. What could be the possible reason behind such a trend?

Reply: Thanks for the remark and pointing out that this might have been a bit unclear in the previous version. We have added clarification of this behavior in the updated text (page 5). The flattening of the resistivity curves and the increased residual resistivity at low temperatures imply a weakening of the electron-phonon coupling and enhanced disorder (both aligning with an enhanced solubility of Ta). In a previous study [*Phys. Rev. B* **103**, 085202 (2021)], we showed that V/Ta substitution results in a softening of the Heusler lattice and a reduction of the electron-phonon deformation potential, thus flattening resistivity curves at elevated temperatures, where acoustic phonon scattering dominates.

Furthermore, the $\text{Bi}_{1-x}\text{Sb}_x$ phase itself has a higher residual resistivity and a flatter resistivity curve than the Heusler compound, thereby also contributing to a flattening of the temperature-dependent resistivity in the composite, when charge transport takes place along parallel transport channels (Heusler grains + BiSb grain boundaries).

9. In the Supplementary Information, authors performed Rietveld refinement for pure and 50% samples and showed variation in the lattice parameters. The authors may like to try to estimate the presence of impurities by adding the secondary phase of BiSb in the refinement process. Similarly, this can be tested for the 20% sample to get the exact trend in variation in the lattice parameters. Additionally, this will also provide the estimation of tentative percentage of secondary phase present in the parent Heusler compound.

Reply: We thank the referee for the suggestion and have included additional Rietveld refinements in the Supplementary Information (Fig. S12). We have added the secondary phase in the refinement process to try and estimate the amount of the secondary phase, as suggested by the referee. However, the peak intensities of the Heusler phase are heavily affected by antisite disorder, which is induced on the surface of the material during hand grinding and other forms of mechanical work – a well known problem in evaluating peak intensities from powder diffraction patterns of Fe_2VAl -based Heusler materials (see e.g. [Acta Materialia 121, 126-135 (2016)] and [Acta Materialia 142, 193-200 (2018)]). Thus, the obtained amount of secondary phase is likely not very reliable and in fact contradicts the values found from EDX mapping analyses (~ 2 vol% $\text{Bi}_{1-x}\text{Sb}_x$ from XRD refinements versus $\sim 5-7$ vol% from SEM and EDX).

10. The interfaces are anomalous and thermodynamically unstable [Nano Letters 12 (8) 4305 (2012)], which affects the overall performance on multiple cycling of the operations. Authors may like to address the interfacial instabilities during operations.

Reply: Thanks for pointing this out. Following the referee's suggestion, we have included measurements for multiple thermal cycles in the Supplementary Information (see Fig. 1 below in this document as well as Fig. S22 in the updated SI). There is no obvious deviation between different, consecutive measurement cycles at least up to $T \sim 500$ K. Thus, we conclude that our samples are thermodynamically stable, at least around room temperature. We expect that the temperature range most relevant for application of these materials would be close to room temperature anyways, given that zT degrades rapidly above 300 K due to bipolar conduction. Thus, we do not expect thermal stability to be a major issue, although we acknowledge that more comprehensive studies would be valuable for a full confirmation. We have added Fig. 1 from this document (Fig. S22) and a short section discussing it to the updated Supplementary Information. We also included a sentence mentioning thermal stability in the main manuscript on page 6: "To investigate the thermal

stability of our composites, transport measurements were conducted for various thermal cycles (Fig. S22), which reveal excellent reproducibility and no degradation of the properties at least up to 500 K – the most relevant temperature range for potential application of these materials.”

11. Authors can also add the designation of symbols used for various compounds in Fig 1a.

Reply: We have added a detailed legend for the symbols in Fig. 1a, as recommended by the reviewer.

Fig. 1: Thermal stability of temperature-dependent thermoelectric properties for the high-performance FVAB50 composite ($\text{Fe}_2\text{V}_{0.95}\text{Ta}_{0.1}\text{Al}_{0.95}$ + 50 vol% $\text{Bi}_{0.9}\text{Sb}_{0.1}$ added before sintering). Various consecutive temperature-dependent measurements yield excellent reproducibility and minimal deviation (within experimental uncertainty) for all investigated thermoelectric properties.

Reviewer #2 (Remarks to the Author):

The study suggests the enhanced carrier mobility by the topologically protected charge transport in the topological insulator $\text{Bi}_{1-x}\text{Sb}_x$ between $\text{Fe}_2\text{V}_{0.95}\text{Ta}_{0.1}\text{Al}_{0.95}$ grains. The topologically protected charge transport is an interesting and important concept for the development of high-performance thermoelectric materials. However, the manuscript does not include reasonable measurement results, making the findings difficult to accept. Therefore, the manuscript is not recommended for publication in this journal due to the following issues.

Reply: We sincerely thank the reviewer for critically evaluating our manuscript. We are pleased that the reviewer finds the topic of topologically protected charge transport in thermoelectric materials of interest. Below, we address each of the reviewer's concerns in detail and provide clarifications to ensure the revised version meets the journal's standards.

1) Firstly, the sample was fabricated using liquid-phase sintering with $\text{Fe}_2\text{V}_{0.95}\text{Ta}_{0.1}\text{Al}_{0.95}$ powder and $\text{Bi}_{0.9}\text{Sb}_{0.1}$. The density of the sample should be provided for sintered samples. Additionally, since the composite consists of different materials, the material ratio can be inferred from SEM/EDX results. The composite material ratio is a key factor in understanding the transport properties of the composite samples.

Reply: We fully agree with the reviewer that the density of the sintered samples should be provided. The reference sample, without any $\text{Bi}_{1-x}\text{Sb}_x$, achieved a density of approximately 95% of its theoretical density. When about 10 vol% of $\text{Bi}_{1-x}\text{Sb}_x$ is added before sintering, the composite material shows minimal porosity. SEM micrographs confirm that any pores present in the initial sample are filled by the secondary $\text{Bi}_{1-x}\text{Sb}_x$ phase, resulting in a density close to 100% for all composite samples. This information has now been added in the updated manuscript (top left paragraph on page 3).

Fig. 2: Microstructure and distribution of secondary phases in $\text{Fe}_2\text{V}_{0.95}\text{Ta}_{0.1}\text{Al}_{0.95}$ reference sample (upper panels) and in the FVAB10 composite sample ($\text{Fe}_2\text{V}_{0.95}\text{Ta}_{0.1}\text{Al}_{0.95} + 10 \text{ vol}\% \text{Bi}_{0.9}\text{Sb}_{0.1}$ added before sintering). The reference sample displays some pores and cavities, which become entirely filled by the liquid BiSb secondary phase during the sintering process.

Regarding the material ratio inferred from EDX/SEM, we have carried out detailed EDX and SEM investigations on several samples and obtained the material ratios, which yield a volume fraction of the $\text{Bi}_{1-x}\text{Sb}_x$ secondary phase of around 2.6 vol% for the FVAB10 composite and 5-7 vol% for the FVAB30 and FVAB50 composites. However, in our opinion, the phase distribution and morphology of the secondary phase network is a much more crucial parameter in determining the thermoelectric properties of the composites. In fact, some of the composite samples display a similar EDX volume fraction but, nonetheless, a different microstructure and different TE properties are obtained. Thus, we think that it is more meaningful and less misleading to differentiate and label the samples by the synthesis conditions rather than the EDX volume fraction.

2) High-performance thermoelectric materials often exhibit anisotropic behavior in sintered samples due to the preferred orientation of their layered structure along the pressing direction. Since $\text{Bi}_{0.9}\text{Sb}_{0.1}$ has a layered structure, the thermoelectric properties should be measured in the same direction to account for the anisotropic behavior of the sintered sample. The authors used a sintered pellet for the LFA measurement and a sample cut from the LFA specimen, meaning that the measurement directions for the ZEM-3 and LFA are perpendicular. Additionally, samples prepared via liquid-phase sintering could be inhomogeneous [Science 348 (2015) 109-114]. To ensure reliable data for the sintered composite, measurements at different sample positions and re-measurements are necessary to confirm homogeneity and thermal stability.

Reply: It is true that single-crystalline $\text{Bi}_{1-x}\text{Sb}_x$ displays anisotropy in its electronic transport properties, although Fe_2VAI (the main phase of our composite samples) is cubic and isotropic even in single crystalline form. The morphology of the $\text{Bi}_{1-x}\text{Sb}_x$ secondary phase is dictated by the structure of the Heusler grains (since this phase only exists between the Heusler grains). However, since the Heusler structure is cubic and no preferential alignment occurs during sintering, the $\text{Bi}_{1-x}\text{Sb}_x$ grain boundary network is also uniformly distributed throughout the sample. Even if the $\text{Bi}_{1-x}\text{Sb}_x$ phase were to crystallize with a preferential alignment along the grain boundaries, the grain boundaries themselves are uniformly and randomly distributed in the sample, that is, there is no slabs/platelet-like geometry of the Heusler grains.

Nevertheless, we in fact measured the thermal conductivity of our composite samples in different directions of the sample (perpendicular and parallel to the compaction direction) and found excellent agreement between the two temperature-dependent measurement curves (see Fig. S21 as well as Fig. 3 in this document below). We had even added a section discussing this in the previously submitted Supplementary Information file (section 7 and Fig. S21), which the referee might have missed. Furthermore, we would like to emphasize

that we have measured our samples in different setups, in different geometries and in different laboratories. We also thoroughly and comprehensively studied the microstructure at various positions to rule out any gradients or inhomogeneities in the composition and distribution of the secondary phases. We also would like to point out that we were aware of the problems associated with the paper [*Science* 348 (2015) 109-114] mentioned by the referee and thus designed our study keeping this in mind.

Regarding sample stability, please confer our response to the comment by the first referee as well as Fig. 1 above in this rebuttal letter.

Fig. 3: (a) Temperature-dependent thermal conductivity of FVAB20 and FVAB50 composite samples ($\text{Fe}_2\text{V}_{0.95}\text{Ta}_{0.1}\text{Al}_{0.95}$ with 20 and 50 vol% $\text{Bi}_{0.9}\text{Sb}_{0.1}$ added during sintering), which were measured in different directions of the sample (parallel and perpendicular to the compaction direction) using different setups as sketched in (b) and (c). Excellent agreement between the different data highlights that there is no significant anisotropy in the microstructure of the investigated samples.

3) The lattice parameters of the FVAB50 composite and $\text{Fe}_2\text{V}_{0.95}\text{Ta}_{0.1}\text{Al}_{0.95}$ differ in Fig. 2(e). This suggests that the thermoelectric properties of the composite may be influenced by doping during the sintering process. Consequently, the enhanced carrier mobility might result from the reduced carrier concentration in the composite.

Reply: We do not think this is the case. In fact, if anything, the increased solubility of Ta in the Heusler structure most likely reduces the carrier mobility due to additional point defect scattering of charge carriers [*Phys. Rev. B* **103**, 085202 (2021)], which is also reflected in a larger residual resistivity ρ_0 of the composite sample at the lowest temperatures. We also do not think that the carrier concentration differs significantly between samples. This becomes apparent when observing the low-temperature Seebeck coefficient. The slope of $S(T)$ at low temperatures is inversely proportional to the Fermi level relative to the band edge. Since all investigated samples display a comparable slope of $S(T)$ at low temperatures, the carrier

concentration should be similar too. Indeed, as obtained from Hall effect measurements, the carrier concentration of pure $\text{Fe}_2\text{V}_{0.95}\text{Ta}_{0.1}\text{Al}_{0.95}$ at 4 K is around $2.0 \times 10^{21} \text{ cm}^{-3}$ and in the FVAB50 composite, we find, from the two-band model, a carrier concentration of $2.1 \times 10^{21} \text{ cm}^{-3}$ for the Heusler phase.

Besides that, the highly anomalous field-dependent Hall effect cannot be explained by a mere change in the carrier doping concentration. On the other hand, we were able to show that it **can** be explained by a two-channel transport model, which considers ultrahigh mobility carriers (orders of magnitude higher than the mobility of charge carriers associated with the Heusler phase) moving along the material's grain boundaries in a topologically protected manner.

4) The composition of the $\text{Bi}_{0.9}\text{Sb}_{0.1}$ may change during the melting process. The electrical properties of the $\text{Bi}_{1-x}\text{Sb}_x$ are significantly influenced by the composition (x) [Journal of Alloys and Compounds 467 (2009) 305–309]. Therefore, since the composition ratio of the $\text{Bi}_{1-x}\text{Sb}_x$ is a crucial factor in this study, the chemical composition should be provided.

Reply: We have also addressed this in the previously submitted version of our manuscript's Supplementary Information and provided the composition of the $\text{Bi}_{1-x}\text{Sb}_x$ phase (see Fig. S11d-f). The composition appears to change during the sintering process when different amounts of the $\text{Bi}_{1-x}\text{Sb}_x$ phase are added. Crucially, however, the thermoelectric performance of our best composite sample at room temperature is significantly larger than the highest room-temperature zT of the whole $\text{Bi}_{1-x}\text{Sb}_x$ system in the entire composition range ($zT_{\text{max}} \sim 0.3$) [Journal of Thermoelectricity 4, 23-36 (2016)]. Therefore, a mere change in the Bi:Sb ratio cannot account for the enhanced thermoelectric performance reported in the present work.

Reviewer #3 (Remarks to the Author):

In this manuscript, a new approach was proposed to decouple the charge and heat transport of Fe₂VAl full-Heusler compounds by incorporating archetypal topological insulator Bi_{1-x}Sb_x. Although the underlying mechanism of the synergistic optimization of transport properties is still unclear, the presented work is interesting and makes a positive contribution to the field of thermoelectricity, in particular the realization of an exciting room-temperature zT for Fe₂VAl alloy. However, before I recommend the publication to Nature Communications, there are some concerns as listed below:

Reply: First and foremost, we want to thank the referee for his time and effort in reviewing our manuscript. We are glad to hear that the reviewer deems our work important and interesting. The replies to the reviewer's comments are given in our point-by-point response below.

1. The effective-medium theory states that zT in composites are always smaller than the largest zT of the individual components. But actually this is not suitable for all many composites, especially for the nanocomposites, for example, BaFe₁₂O₁₉/Ba_{0.3}In_{0.3}Co₄Sb₁₂ in Nat Nanotechnol 2017, 12, 55 and (Fe,Co,Ni)/Ba_{0.3}In_{0.3}Co₄Sb₁₂ in Nature 2017, 549, 247. zT values for the nanocomposites are apparently higher than those of the individual components. Thus, the exceeding the predictions of effective-medium theory, as claimed by authors should be a common phenomenon for composite materials.

Reply: We appreciate the reviewer's comment on the limitations of effective-medium theory in predicting the thermoelectric performance of composite materials, particularly in the context of nanocomposites. We are aware that the TE properties in nanocomposites can often deviate from effective medium theory predictions. Of course, reducing lattice thermal conductivity via engineering nanostructures in composite materials has been one of the most successful strategies to enhance performance. At the same time, however, electrical conductivity is often reduced owing to charge carriers scattering off heterogeneous interfaces. Lastly, it is well known that even the Seebeck coefficient can be improved in nanocomposites due to energy filtering. Still, we were quite surprised that in the composites from the present study, which display microscale (rather than nanoscale) structures, huge deviations from effective medium theory were realized. We have added relevant references as suggested by the referee to the updated manuscript and adjusted the text accordingly to point out that deviations from effective medium theory have been reported previously (see the added red text at the top left of page 6).

2. Long-term stability and reliability are crucial for the practical application of thermoelectric materials. The assessment of the stability and reliability of the composite material over

extended periods is lacking. I suggest that the authors supplement the manuscript with long-term thermoelectric performance data, including but not limited to changes in microstructure thermal conductivity, electrical conductivity, and Seebeck coefficient over time.

Reply: We have now added in the updated Supplementary Information a discussion of the thermal stability of our samples (see also Fig. 1 above as well as our reply to the first reviewer). There is no significant deviation in the temperature-dependent physical properties of our high-performance composite sample, at least when measuring up to 500 K for several consecutive thermal cycles. Please note that the optimal performance of these composites lies around 300 K (or even slightly below) and that application of these materials at much higher temperatures is therefore not desirable anyways.

3. The author pointed out that by incorporating archetypal topological insulator $\text{Bi}_{1-x}\text{Sb}_x$ between $\text{Fe}_2\text{V}_{1-x}\text{Ta}_x\text{Al}$ grains, the solid solubility of Ta is also increased. However, it remains to be further proved whether the enhanced thermoelectric performance is dominated by the two-channel transport model of the composite material as proposed in manuscript, or the enhanced doping effect caused by Ta. So, the influence and contribution of Ta doping to the thermoelectric properties should be detailed discussed.

Reply: This is an excellent question raised by the referee and we have added a more detailed discussion of the effect of Ta doping on the thermoelectric properties in the updated manuscript (page 5). However, despite comprehensive investigations, it is difficult to pinpoint whether the change in TE properties is being dominated by the increased Ta solubility or by the two-phase microstructure, and further studies are required. For instance, changing the type of topological-insulating secondary phase could be explored and another interesting direction to explore would be to increase the volume fraction of the secondary phase by reducing the grain size of the Heusler material. These endeavors, however, go well beyond the scope of the present work, which is already quite extensive.

However, given that the maximum zT of Ta-substituted Fe_2VAl Heusler alloys does not exceed 0.2-0.3 ([*Journal of Appl. Phys.* **115** (2014)], [*Acta Materialia* **212**, 116867 (2021)]), we are convinced that the topological-insulating grain boundary network plays a significant role, particularly in retaining high carrier mobilities within the composite sample, as evidenced by the anomalous Hall response.

4. The paper utilizes the two-band model to calculate the temperature-dependent Seebeck coefficient for $\text{Fe}_2\text{V}_{0.95}\text{Ta}_{0.1}\text{Al}_{0.95}$ and $\text{Bi}_{0.9}\text{Sb}_{0.1}$, with experimental values being consistent with theoretical values. However, the band structure information for these materials is missing. I suggest that the authors provide the band structure information for

these two materials. Furthermore, Fig. S9 indicates a significant fluctuation in the ratio of Bi to Sb at the grain boundaries. At this point, the comparison of the Seebeck coefficient calculated using the two-band model for Bi_{0.9}Sb_{0.1} with experimental values seems to be no reliability.

Reply: We modelled the temperature-dependent Seebeck coefficient of the pristine materials, Fe₂V_{0.95}Ta_{0.1}Al_{0.95} and Bi_{0.9}Sb_{0.1}, using a two- and triple-parabolic band model, respectively. The extracted information for the effective band structures along with a sketch of the bands and relevant energy gaps are shown in Fig. S15b. As the reviewer rightly pointed out, discussing the properties of the composite solely based on the individual characteristics of Fe₂V_{0.95}Ta_{0.1}Al_{0.95} and Bi_{0.9}Sb_{0.1} is challenging and most likely wrong, given the changes in both the Heusler phase (due to increased Ta substitution) and the Bi_{1-x}Sb_x phase (with fluctuations in the Bi:Sb ratio). Therefore, we believe that focusing too heavily on the details of the electronic band structure of the individual components is not particularly meaningful in this context. Instead, the detailed analyses of the individual materials were moved to the Supplementary Information for the interested reader. We would like to emphasize, however, that despite the compositional fluctuations in the Bi_{1-x}Sb_x phase, the FVAB50 composite, whose magneto-transport properties are extensively discussed in the manuscript, remains within the topological-insulator regime (see Fig. 3 below). This regime spans a broad composition range of $0.07 < x < 0.22$ in Bi_{1-x}Sb_x. Our modelling of the field-dependent Hall effect in the manuscript does not require any preliminary knowledge of the two materials, yet the fits reveal an ultrahigh-mobility channel, consistent with the Dirac-like surface states of Bi_{1-x}Sb_x ($0.07 < x < 0.22$), and a lower mobility channel consistent with the bulk values of heavily doped Fe₂VAL-type Heusler compounds.

Fig. 3: Sketch of the electronic structure phase diagram of the Bi-Sb system. The material undergoes two transitions from a topological semimetal towards a topological insulator and vice versa at $x \sim 0.07$ and 0.22 , respectively. Figure was reprinted from [Journal of Thermoelectricity 4, 23-36 (2016)].

Response to Review (NCOMMS-24-72976-T)

Reviewer #1 (Remarks to the Author):

The authors have incorporated the reviewers' suggestions and addressed their concerns satisfactorily. The revised version is now suitable for consideration.

Reply: We thank the referee for reviewing our updated manuscript and appreciate that it can now be recommended for publication.

Reviewer #2 (Remarks to the Author):

In my opinion, the manuscript has been improved and is acceptable after the corrections. Therefore, I recommend its publication in Nature Communications.

Reply: We also thank the second referee for reviewing our updated manuscript and rebuttal letter and are pleased to hear that the updated manuscript can be recommended for publication in Nature Communications.